# HANDSON Hand: Strategies and Approaches for Competitive Success at CYBATHLON 2024

**DOI:** 10.3390/bioengineering12030228

**Published:** 2025-02-24

**Authors:** Xuhui Hu, Fengkai Guo, Zhikai Wei, Dapeng Chen, Junfa Dai, Anran Li, Senhao Zhang, Mostafa Orban, Yao Tong, Cong Hu, Baoguo Xu, Hong Zeng, Aiguo Song, Kai Guo, Hongbo Yang

**Affiliations:** 1Suzhou Institute of Biomedical Engineering and Technology, Chinese Academy of Sciences, Suzhou 215163, China; michael.xuhui.hu@outlook.com (X.H.); d_junfa@foxmail.com (J.D.); mustafa.essam@feng.bu.edu.eg (M.O.); ty1390@mail.ustc.edu.cn (Y.T.); guok@sibet.ac.cn (K.G.); 2School of Instrument Science and Engineering, Southeast University, Nanjing 210096, China; teaser_guo@163.com (F.G.); 230228933@seu.edu.cn (Z.W.); xubaoguo@seu.edu.cn (B.X.); hzeng@seu.edu.cn (H.Z.); 3Guangxi Key Laboratory of Automatic Detecting Technology and Instruments, Guilin University of Electronic Technology, Guilin 541004, China; hucong@guet.edu.cn; 4School of Automation, Nanjing University of Information Science and Technology, Nanjing 210044, China; dpchen@nuist.edu.cn; 5School of Electrical Engineering and Automation, Shandong University of Science and Technology, Qingdao 266590, China; anrannarl@hotmail.com; 6Mechanical Department, Faculty of Engineering at Shoubra, Benha University, Cairo 11511, Egypt; 7Shenzhen Research Institute, Southeast University, Shenzhen 518055, China

**Keywords:** assistive devices, prosthetic hand, CYBATHLON

## Abstract

A significant number of people with disabilities rely on assistive devices, yet these technologies often face limitations, including restricted functionality, inadequate user-centered design, and a lack of standardized evaluation metrics. While upper-limb prosthetics remain a key research focus, existing commercial solutions still fall short of meeting daily reliability and usability needs, leading to high abandonment rates. CYBATHLON integrates assistive technologies into daily living tasks, driving innovation and prioritizing user needs. In CYBATHLON 2024, the HANDSON hand secured first place in the arm prosthesis race, showcasing breakthroughs in human–robot integration. This paper presents the HANDSON hand’s design, core technologies, training strategies, and competition performance, offering insights for advancing multifunctional prosthetic hands to tackle real-world challenges.

## 1. Introduction

Millions of individuals worldwide rely on assistive devices. However, the practical application of such technologies is often hindered by numerous limitations. For example, wheelchairs struggle with stair navigation [1,2], prosthetic limbs face restricted sensory and motor capabilities [3,4,5], and assistive technologies for the visually impaired remain inadequate [6,7]. Furthermore, insufficient communication among developers, individuals with disabilities, therapists, and clinicians frequently results in devices that fail to align with user needs [8]. The objective evaluation of assistive technologies presents an additional challenge, as no standardized metrics currently exist to comprehensively assess their effectiveness [9,10]. These combined factors contribute to the rejection or suboptimal utilization of many assistive devices [11]. For instance, among upper-limb prosthesis users, approximately one-quarter discontinue use, citing issues such as discomfort, high cost, aesthetic concerns, and functional limitations [9].

Nevertheless, the introduction of competitive challenges focused on assistive technologies presents a promising path to tackle these issues. A notable example is CYBATHLON, initiated by ETH Zurich, which originates from the combination of “Cyber” and “Athlon”. Unlike the Paralympic Games, it integrates advanced assistive devices into tasks simulating activities of daily living (ADLs) [12]. This competition highlights the interplay between individual capabilities and cutting-edge technologies. To succeed, research teams are incentivized to advance technological innovation while prioritizing user needs. Additionally, such competitions provide an equitable platform to assess the effectiveness of diverse technological approaches in enhancing the quality of life for individuals with disabilities.

The loss of upper-limb functionality profoundly affects various dimensions of daily living. Current prosthetic solutions are categorized into cosmetic, body-powered (BP) [13,14], and motor-powered (primarily using myoelectric control methods, MP) prostheses [9,15,16]. Each type has distinct advantages and limitations. Cosmetic prostheses primarily serve aesthetic purposes, offering limited functional utility but contributing to psychological well-being, particularly in social interactions. BP prostheses, while effective for heavy-load tasks, are functionally constrained. In contrast, MP prostheses, supported by recent advancements in robotics and artificial intelligence [17,18,19,20,21], demonstrate the highest potential for restoring fine motor skills [19,22], particularly in low-intensity tasks [23,24,25]. As a result, contemporary research predominantly focuses on MP systems [26,27,28]. Nevertheless, no current prosthetic solution adequately addresses all user needs comprehensively [29,30,31].

Since its inception, CYBATHLON has featured the “Arm Prosthesis Race” (ARM Discipline), which aims to evaluate and showcase the comprehensive capabilities of prosthetic arms, including heavy-load lifting, multijoint dexterous control, and sensory feedback integration. To accommodate advancements in assistive technology, the tasks in each competition have become progressively more challenging. Nonetheless, the competition consistently focuses on core technical aspects, such as heavy lifting, fine manipulation, bi-manual coordination, operation within large work spaces, wrist control, and sensory feedback.

Interestingly, despite the increasing complexity of tasks and the predominance of MP-based solutions among participating teams, the champions of the 2016 and 2020 competitions were BP teams (Team DIPO Power, 2016 and Team Makerhand, 2020). BP prostheses, though limited to basic open-and-close functions, excelled because of their superior reliability, lightweight design, and simple control mechanisms. These advantages allowed pilots to operate the devices with greater ease and adaptability. Additionally, by using nonphysiological compensatory movements from the torso, the prostheses’ terminal devices could perform actions equivalent to wrist flexion or rotation. This enabled the pilots to successfully tackle the challenging tasks in the competition. Although the competition allows the use of certain body-compensation “tricks”, the primary goal for both organizers and participating teams is to promote the development of advanced prosthetic technologies. For instance, the 2024 competition introduced new rules that restricted usage of compensatory motion and increased task difficulty, placing greater emphasis on the prosthetic hand’s capabilities, such as strong grip force and force-tactile perception (details provided in Section 2). These challenges pose significant obstacles for both body-powered (BP) and motor-powered (MP) prostheses, as no existing commercial or research-based products are currently able to fully meet all the competition’s tasks.

The HANDSON hand was mainly developed by a collaborative research team from Southeast University and the Suzhou Institute of Biomedical Engineering and Technology (SIBET), Chinese Academy of Sciences. Some team members previously participated in CYBATHLON 2020 as Team Manager and Pilot under the name “Team Hands On”, achieving 8th place. The authors of this paper personally participated in the online challenges held in March 2023 and February 2024, competing as Team SIBET and achieving first and third places, respectively. Most recently, Team HANDSON competed in the CYBATHLON 2024 finals, held in Zurich, Switzerland, in October 2024 (90 points, 1st place). This paper focuses on the efforts made by the HANDSON team in CYBATHLON 2024. It details the pilot’s background, the key technologies employed in the prosthetic device, the training strategies, and the final performance. The findings aim to contribute to the exploration of future prosthetic hand designs with enhanced multifunctionality and to validate their feasibility and usability in addressing real-world challenges.

## 2. Materials and Methods

### 2.1. Pilot Information

The subject, Min Xu, female, born in November 1975, underwent a unilateral (right), transradial amputation. The amputation occurred in 1992, when the subject was at the age of 17, as a result of a factory punching machine accident (X-ray images provided in Figure 1). Before the incident, she was right-hand dominant. In 2000, the subject briefly used a cosmetic prosthesis for her wedding. However, she quickly abandoned it because of discomfort, specifically finding the socket unbearable hot and stifling. She did not use any prosthetic devices again until January 2023, when she began training for the CYBATHLON 2023 Challenge. This marked her first experience with functional prostheses. As a pilot, she successfully participated in the CYBATHLON Challenge events in March 2023 and February 2024, as well as CYBATHLON 2024. Prior to her participation in prototype testing, the training was approved by the regional Ethics Committee, and she signed an informed consent (Appendix A). The ethical approval and device risk analysis was reviewed by the organizing committee of the CYBATHLON before the subject was granted permission to participate in competition.

### 2.2. Rules in CYBATHLON ARM 2024

The ARM discipline in CYBATHLON 2024 consisted of two stages: the qualification round and the final round. Before the qualification round, teams could choose to participate in rehearsals and training sessions. During the qualification round, each pilot had two attempts, with performance evaluated based on the number of tasks completed and the time taken. The top four pilots from the qualification round advanced to the final to compete for the championship titles.

According to the rules, certain tools used in the competition were marked in blue, indicating that they must be manipulated exclusively with the prosthetic device. This rule minimized reliance on the intact hand, placing greater emphasis on the functionality of the prosthesis and its practical utility. In CYBATHLON 2024, pilots were asked to complete 10 daily life-inspired tasks, including heavy lifting, bimanual coordination, fine object manipulation, and tactile perception, within an 8 min time limit. Any violation during a task resulted in an immediate “0 point” for that task, imposing strict demands on both the precision and speed of the pilots’ performance.

For detailed official rules, please refer to Appendix A. This paper highlights a few key changes compared with the 2020 competition, as illustrated in Table 1. In terms of general rules: (a) To encourage teams to enhance the versatility of prosthetic hand grasping functions, midcompetition replacement of prosthetic components was prohibited. This rule prevented teams from using different prosthetic components tailored to specific tasks. (b) To promote the reliability of prosthetic control, teams were forbidden from touching the prosthetic hand with other body parts while it was gripping an object. This measure prevented the use of temporary power cuts to maintain a grip position, which could artificially enhance load capacity for lifting heavier objects.

For specific task rules, the number of tasks increased from 6 to 10 compared with the 2020 competition, and the difficulty of the original 6 tasks was further elevated. As a result, both the number and complexity of the tasks were significantly increased, while the 8 min time limit remained unchanged. The 10 tasks were as follows:***(New Task)* *Carrying Bottles*:** A series of bottles of different weights must be placed in a bottle crate, and the crate must then be carried to a shelf; the bottles must then be removed from the crate and placed on top of the shelf.***(New Task) Serving Food*:** A casserole dish and a frying pan must be carried from the stove to a predefined location on a table.***(New Task) Storing Dishes*:** Typical kitchen objects must be grasped and stowed away at predefined target locations.***Hanging Laundry*:** Pilots must put on a hooded sweater, fully close the zipper, take it off, and hang it over the clothesline. Finally, the pilot must hang up a t-shirt on the clothesline using a blue clothespin. In 2024, the task added difficulty by requiring both the zipper and clothespin to be operated exclusively with the prosthetic hand. Additionally, the clothesline was raised to a greater height, further increasing the task’s complexity.***Do-it-yourself (DIY)*:** Pilots must use a variety of hand tools in the context of DIY-type work. In 2024, the task included using pliers to remove a nail and screwing a blue light bulb into a holder situated in a confined space, making it significantly harder to position and secure the bulb.***Containers*:** Pilots must conduct a series of container-related bimanual tasks.***Haptic Bag*:** Pilots must identify and handle items inside a soft bag without visual cues. The 2024 competition emphasized advanced haptic feedback and fine sensation, necessitating the integration of sensors to recognize both the shape and texture of objects.***(New Task)*** ***Hot Wire*:** Pilots hold a conductive wire loop with a blue handle. A curved metal wire must be tracked without touching the wire with the loop by using the prosthetic hand only.***Stacking*:** Pilots sit in front of a table and must stack blue cups into a vertical pyramid. The 2024 version prohibited body compensations, requiring the task to be performed entirely while seated.***Clean Sweep*:** Pilots are asked to grasp and move blue objects individually with their prosthetic hand from their random, initial position on a table surface to a target position on a neighboring table.

### 2.3. Prosthetic Device

To overcome these challenges, it was essential to address numerous technical difficulties related to the prosthetic hand while also incorporating user-centered design principles. This ensured that the pilot could minimize reliance on rule-permitted tricks during the competition. Specifically, we established the following technical criteria to enhance the pilot’s performance: **(a) Prosthesis Weight Under 1 kg:** Reducing the weight minimized user fatigue and helped prevent misoperations caused by changes in surface electromyograph (sEMG) signal characteristics. Moreover, lower inertia in the prosthetic device enabled smoother and more precise control during operation. **(b) Active Wrist Rotation and Passive Wrist Flexion:** These features addressed tasks requiring wrist mobility, such as twisting a light bulb or stacking cups. **(c) Maximum Grip Force > 80 N:** Ensured that the prosthesis could handle high-load tasks in the competition, such as hammering nails or pulling out pins. **(d) Object Shape/Stiffness Recognition Rate ≥ 85%:** Enabled effective completion of the “Haptic Bag” task.

The system framework of the HANDSON hand is shown in Figure 2. The hand integrated multiple functional modules to achieve precise control and adaptability. The prosthetic hand and wrist, powered by servo motors, served as the main body of the system, enabling reliable and precise movements. As our technical solution required controlling two active joints, achieving high reliability for both joints through single EMG control was more challenging. Inspired by body-driven mechanisms, we designed a dual-source control strategy in our solution: leveraging the body-control source, which serves as a simpler and more reliable control signal, to map to controlling the opening and closing of the prosthetic hand, while the EMG signal was responsible for controlling wrist rotation. As a result, a wearable shoulder pad equipped with stretch sensors worked in conjunction with the body-driven controller to monitor shoulder movements, translating them into commands for controlling the prosthetic hand’s opening and closing. The myoelectric control system used electrodes placed on the residual limb to capture muscle activation signals, which were processed by the main control unit (STM32U5 series microcontroller, STMicroelectronics Inc. Geneva, Switzerland) to enable proportional control of wrist rotation. The main control unit acted as the central hub, integrating data from the motion-sensing shoulder pad and electrodes while communicating with other components, such as the motor controller and the expansion board. The system also included a vision module, consisting of a camera and an AI edge-computing module (MaixCAM, Sipeed Inc. Shenzhen, China), which handled visual information to assist with object recognition. For human–machine interaction, LED buttons and a buzzer provided intuitive feedback and ensured fundamental communication between the pilot and the system. Additionally, the motor controller, based on an Arduino Leonardo board, managed the servo motors for hand and wrist movements, ensuring precise responses and force regulation. Finally, the battery module, a 3200 mAh rechargeable lithium battery, provided power to the entire system, ensuring up to 12 h of smooth and reliable operation. This integrated and modular design enhanced the prosthetic’s functionality, control reliability, and overall user experience (detailed specifications can be found in Appendix A).

To further reduce the weight of the prosthetic hand, only the transmission components and force-application parts of the gripper were made from metal. Other components, such as the prosthetic socket and wrist joint, were digitally modeled and fabricated using PLA material via 3D printing. The total wearable weight was reduced to 900 g, significantly decreasing user fatigue and improving control reliability. The design also accounted for proper cable management and ensured compliance with safety regulations to meet competition requirements. Figure 3 shows the relationship among these key points. The specific methods behind these contributions are detailed in the next subsection. Compared with existing commercial prosthetic products and other laboratory prototypes from competing teams, our innovations and contributions focus on three key areas:Reliable Multi-DOF Prosthetic Hand Functionality: The HANDSON hand was able to complete all tasks specified in the competition, demonstrating exceptional functional performance.Novel Intention-Recognition Method: This approach reliably controlled the proportional synchronous motion of two degrees of freedom, hand grasping and wrist movement.Integrated Shape-/Stiffness-Recognition System: Built into the prosthetic device, this system effectively provided feedback to the pilot, assisting in the accurate grasping of target objects in the “Haptic Bag” task.

#### 2.3.1. Mechanical Design of the Hand and Wrist

The prosthetic hand was designed with a series of degrees of freedom (DOFs) arranged sequentially from proximal to distal: wrist flexion, wrist rotation, and finger opening/closing. **(a) Wrist Flexion DOF:** Controlled manually using a mechanical trigger, allowing the wrist to be locked at either 0° or 90°. **(b) Wrist Rotation DOF:** Actuated by a servo motor (4.5 N.m, TIANKONGRC Inc.), this function enabled tasks such as stacking objects into a pyramid shape. It could also work in conjunction with the wrist flexion joint to perform operations such as twisting a light bulb in confined spaces. **(c) Finger Opening/Closing DOF:** The prosthetic hand was designed as a single-DOF two-finger gripper driven by a servo motor (4 N.m, SM40BL, FEETECH Inc.). Finger movements were achieved through a parallel linkage mechanism, providing reliable grip functionality.

To ensure the prosthetic hand could handle objects of varying sizes and stiffness (e.g., cards, marbles, light bulbs, and water bottles), the team optimized the design by functionally segmenting the grasping regions of the fingers, as shown in Figure 4a. The middle phalanx was designed for gripping heavy objects, which helped reduce power consumption during finger locking. The proximal region of the distal phalanx featured a concave arc to increase the contact area, enhancing stability when holding tools such as hammers and pliers. The distal region of the distal phalanx was coated with a silicone flat surface to facilitate secure pinching of small objects. Additionally, a closed-loop current-position control system enabled precise force regulation at the fingertip, allowing the prosthetic hand to autonomously maintain a constant grasping force threshold during the competition, thereby reducing the user’s cognitive load. The silicone gripping surfaces distributed pressure evenly to prevent damage to fragile objects, such as light bulbs, while simultaneously enhancing friction to secure heavier items, such as hammers, ensuring stability and preventing slippage. This design effectively balanced the requirements for precision and robustness across diverse tasks.

#### 2.3.2. Intention Recognition Methods

During the competition, the pilot needed to frequently control multiple degrees of freedom of the prosthetic hand. However, because of the limited control signals available from the residual limb, ensuring reliable control of both the fingers and wrist became a priority. Inspired by body-powered prosthetic technologies, we electrified the traditional mechanical cable-pulling structure by developing a wearable and flexible motion-sensing shoulder pad. This pad, embedded with stretch sensors (ESSA series, ElasTech Inc., Carmi, IL, USA), monitored the pilot’s shoulder movements. The greater the shoulder displacement, the wider the finger opening, and vice versa. Thus, it was used in competition as a voluntary-open device, maintaining functional consistency with conventional body-powered prostheses. Additionally, a pair of commercial sEMG electrodes (13E200 = 50, Ottobock Inc., Hong Kong, China) were installed within the prosthetic socket, positioned over the flexor and extensor muscle sites of the residual limb. These electrodes enabled proportional control of wrist rotation in two directions. By combining shoulder movements with sEMG signals from residual limb muscles, the pilot quickly learned to simultaneously control the finger opening angle and wrist rotation, improving overall performance during the competition.

#### 2.3.3. Haptic Feedback for Shape and Material

To address the bottleneck of shape material perception in the “Haptic Bag” task, the team ultimately chose a machine vision solution. In terms of hardware, the team integrated a camera (DECXIN-V3, DCX Inc., Shenzhen, China) at the base of the fingers, with additional lighting around the camera that could be manually switched on and off. On the software side, the team deployed the yolov5 image detection model in the edge computing module (MaixCAM) to identify four types of objects with different shapes and materials as specified in the task. In this task, the prosthetic limb used machine vision recognition technology to convert the image information from the camera into haptic signals that were fed back to the pilot, assisting the pilot in completing the grasping task. To train the required perception model, we collected and annotated over 5000 training set images, taking into account multiple shooting angles, different shooting distances, and various object stacking conditions.

To receive correct feedback on the recognition results and grasping status, the pilot needed to insert the prosthetic hand vertically into the blind box and turn on the camera, using the trained image detection model to detect the four types of objects in the box. Four buttons with LED indicators were installed on the socket surface of the prosthetic limb, and the input functions of the four buttons were mapped to the recognition of the “four types of objects” (when the pilot pressed the button labeled “hard cube”, they input the command to recognize the corresponding object). The visual cues from the four LEDs represent four navigation directions (when the LED indicating “up” lights up, it means the pilot needs to move the gripper in the corresponding direction). Finally, when the target object is within the area that the gripper can grasp, a buzzer alerts the pilot to control the hand to close. With the above feedback information, the user could then grasp the object with the specified shape.

### 2.4. Progress of Technology Readiness Level (TRL)

The CYBATHLON 2024 competition content was first announced in November 2021. While minor adjustments were made to the rules over time, the overall tasks remained consistent, giving teams nearly two years to refine their designs and validate their approaches. Technology Readiness Levels (TRL) objectively reflect the maturity of assistive technologies [32]. The prosthetic device described in this paper has reached a TRL of 7 by now. It was at TRL 3 when development began. This section outlines the team’s progress, mapped against TRL stages.

***TRL 3—COMPONENT EVALUATION (October 2022 to February 2024)*:** In October 2022, the team began preparations by familiarizing themselves with the competition process, tasks, and rules. By January 2023, the pilot was selected and registered under the SIBET Team. At this stage, the team developed a basic set of design requirements based on the announced competition tasks (***TRL 1—BASIC PRINCIPLES***). Before CYBATHLON 2024, the team participated in two annual challenge events as SIBET. The March 2023 challenge tested tasks 1 and 10 (as shown in Figure 5a), while the February 2024 challenge included tasks 2, 4, 5, and 9 (as shown in Figure 5b). During this period, the formulated concept and application were refined through multiple iterations (***TRL 2—TECHNOLOGY FORMULATION***). The hand structure and control logic underwent significant updates. After completing the two challenge events, the team finalized an experimental proof of concept for CYBATHLON 2024.

***TRL 4—OFFLINE PROTOTYPE (May to July 2024)*:** During this phase, the research team developed a functional prototype and tested its components in a controlled lab environment. Monthly sessions with the pilot were conducted to validate the feasibility of human–device integration and gather feedback for refinement.

***TRL 5—ONLINE PROTOTYPE (August 2024)*:** The team focused on improving the practicality of the device, with biweekly sessions with the pilot to test the technology in simulated competition environments. This ensured the prosthetic hand could support the pilot in sequentially completing all 10 tasks and familiarized the pilot with the race structure.

***TRL 6—PROTOTYPE SYSTEM (September 2024)*:** Weekly training sessions lasting approximately two hours were held to enhance the pilot’s proficiency in individual tasks, improving both success rates and task completion times. The research team made adjustments to the prosthetic structure and control logic based on pilot feedback. At this stage, the technology was effectively demonstrated and validated in the competition environment (as shown in Figure 5c).

***TRL 7—DEMONSTRATION SYSTEM (October 2024)*:** With the system’s functionality and control logic finalized, the focus shifted to maintaining device reliability and mitigating potential instabilities. Training frequency increased to 2–3 sessions per week, each lasting 3–4 h. These intensive sessions aimed to enhance the pilot’s task fluency and improve reliability in error-prone tasks. (The final training video can be found in Appendix A.)

## 3. Results

### 3.1. Testing in the Lab

We established a simulated competition environment at the pilot’s home to conduct testing. Leveraging our prior participation in two CYBATHLON Challenges, we already possessed official infrastructure for tasks 1, 2, 4, 5, 9, and 10. The remaining infrastructure was procured mainly from IKEA, as officially designated, while some of the items for the “Haptic Bag” and “Hot Wire” tasks were assembled using self-purchased materials. To minimize the potential impact of infrastructure discrepancies on final performance, the team traveled to Zurich at the end of August 2024, carrying the prosthetic prototype for on-site training. Since the pilot did not accompany the team, the focus was primarily on evaluating the offline performance of the prosthetic hand.

Figure 6 illustrates the progressive improvement in task completion times, reflecting the enhanced human–machine integration achieved through cumulative training. The overall trend reveals significant reductions in completion times, demonstrating the positive impact of sustained training. Notably, the “Haptic Bag” task was introduced later, starting on 7 October, with prior tests recorded as zero seconds. Complex tasks such as “Hanging Laundry” and “Clean Sweep” initially required more time but exhibited marked improvement in subsequent trials. Similarly to the R&D curve of SoftHand Pro reported in [10], this analysis underscores the evolving synergy between prosthetic technology and user capability, showcasing the potential of iterative training in optimizing performance.

Among the tasks, “Clean Sweep”, “Haptic Bag”, and “Hanging Laundry” were the most time-consuming and the least consistent for the team. The primary sources of instability included marbles potentially slipping off the holder during “Clean Sweep”, severe visual occlusion of target objects inside the blind box during “Haptic Bag”, and the zipper potentially slipping out of the user’s grip during “Hanging Laundry”. These challenges required the participant to develop finer coordination with the prosthetic device through regular and meticulous training sessions.

Figure 7 illustrates the dual objectives of the training process: reducing task error rates to improve scores and minimizing task completion times to meet the 8 min (480 s) race time limit, as indicated by the blue dashed line. During the initial training stages (27 August to 7 October), the primary focus was on completing a full race run, with a higher priority given to reducing error rates, resulting in significant score improvements despite task completion times exceeding the time limit. After 7 October, with the introduction of the “Haptic Bag” task, the focus shifted to optimizing task completion times, ensuring sufficient time allocation for more time-intensive and less stable tasks such as “Haptic Bag” and “Hanging Laundry”. By 20 October, the team generally achieved a balance between high scores and task completion times close to the 480 s threshold. This progression highlights the strategic shift in training focus, effectively balancing accuracy and efficiency to optimize overall performance.

### 3.2. Performance in CYBATHLON 2024

The “Arm Prosthesis Race” (ARM) has consistently been one of the most competitive disciplines in the CYBATHLON competition. In 2024, 18 teams from nine countries registered for the event. Ultimately, 12 teams completed the competition (2 teams registered but did not participate, and 4 teams participated but did not finish any tasks, marked as DNF). The average score for all teams that completed the competition (excluding DNF teams) was 50 points, with six teams scoring above the average.

Our team’s competition schedule was concentrated from the 24 October to the 26th, with the 24th designated as rehearsal day, the 25th for qualification rounds, and the 26th for the finals. Figure 8 illustrates our team’s performance during rehearsal day and the qualification rounds (there were two attempts in the qualification rounds: Qualifications 1 and Qualifications 2). Before the rehearsal and qualification rounds, each team had 30 min and 1 h, respectively, to train on the competition field and test the prosthetic arms.

During the prerehearsal training session, we discovered that the clothespins used in the “Hanging Laundry” task were different from those used in the laboratory and were prone to slippage. Because of the tight schedule between training and rehearsal, we were unable to make timely adjustments to address this issue. As a result, during the rehearsal, the pilot had to abandon the task after multiple failed attempts to grasp the clothespins, which was recorded as 0 points. In the subsequent “Hot Wire” task, the pilot accidentally touched the handle of the loop (blue part) with the contralateral hand, resulting in 0 points. Furthermore, during the “Stacking” task, the pilot accidentally knocked over a cup, leading to another 0 points. Finally, the pilot completed the last task but exceeded the time limit (also 0 points) because of excessive time spent on earlier ones, scoring only 60 points.

Following the rehearsal, the team worked to adjust the grip strength and angle of the hand prosthesis, resolving the issue with grasping the clothespins. However, during the Qualifications 1 round the next day, the pilot was penalized for manipulating the zipper in the “Hanging Laundry” task and finished with a score of 90 points. In the subsequent Qualifications 2 round, the total time taken exceeded 480 s, resulting in 0 points for the final task. Ultimately, HANDSON advanced to the finals with a score of 90 points, securing second place in the group stage.

Figure 9 presents snapshots of the pilot from Team HANDSON completing various tasks during the CYBATHLON in Zurich. HANDSON, equipped with sensory feedback capabilities, was the only team in this year’s competition able to complete the “Haptic Bag” task. By the morning of 26 October 2024, all qualification rounds had been completed. Team Bionicohand made a strong comeback, also achieving a total score of 90 points (skipping only the “Haptic Bag” task). However, they surpassed HANDSON by completing the tasks in less time, securing the top position in the qualifications. Consequently, Bionicohand, HANDSON, BionIT Labs, and MiaHand advanced to the finals. In the afternoon of 26 October 2024, the finals began. MiaHand, ranked fourth, started first in lane one, with the other teams starting in ascending rank order at one-minute intervals. Figure 10 illustrates the task completion times and score progression of the four teams in the finals.

During the finals, our pilot successfully completed the typically time-intensive “Hanging Laundry” task early on, allowing more time for subsequent tasks. However, during the “Haptic Bag” task, the randomized arrangement of objects inside the blind box posed a challenge. Sometimes, the target object was buried deeper within the box under other items, requiring extra time for identification and retrieval. Given Bionicohand’s superior performance in completing other tasks more quickly, the predetermined strategy of team HANDSON was to achieve maximum points by completing all tasks. Despite the delay, our pilot patiently completed the “Haptic Bag” task. At this point, only 150 s remained with three tasks still to be completed. Based on prior training outcomes, the pilot was still projected to have a high chance of achieving a full score within the remaining time.

In the final stages of the competition, Bionicohand lost points in Tasks 7 and 8, allowing our team to narrow the score gap to 20 points after successfully completing the “Hot Wire” task. However, an error is made by our pilot in the subsequent “Stacking” task reduced the score gap to just 10 points. With 70 s remaining, the competition came down to the final task, where no further mistakes could be afforded. Under immense pressure, our pilot completed the final “Clean Sweep” task with just 10 s left on the clock, successfully positioning the last marble and finishing the race. HANDSON ultimately secured the championship with a total score of 90 points.

## 4. Discussion

At CYBATHLON 2024, numerous advancements in prosthetic technology were showcased, with several teams presenting innovative solutions that contributed to their success. Our team’s gripper-based design, while less dexterous than five-fingered bionic hands, demonstrated strong compatibility across various grasping modes and maintained competitive performance in precision tasks. By integrating wrist rotation and flipping capabilities within strict spatial and weight constraints, our device enabled faster task completion in activities such as rotating light bulbs and stacking cups. Notably, we achieved a breakthrough in the “Haptic Bag” task, not only being the first team to successfully tackle this task but laying a crucial hardware foundation for future advancements in embodied intelligence for prosthetic hands.

However, multijoint active control in prosthetic devices does not necessarily translate to enhanced dexterity for users. Multijoint simultaneous control based on myoelectric signals remains an area requiring deeper research, particularly in improving reliability [15,33,34]. Combining traditional cable-driven mechanisms with myoelectric interfaces as a supplementary control source is an intuitive idea [10]. However, traditional cable-driven methods are constrained by the rigidity of mechanical connections. For instance, when the shoulder is already in an extended range of motion (e.g., during Task 2, retrieving the casserole dish from the oven, which requires bending forward and reaching into the oven), the mechanical hand may unintentionally open and struggle to close. The result of this paper indicated that electrification of a cable-driven system improved its performance limits, demonstrating better reliability and practicality across various tasks.

Other teams also showcased remarkable innovations. For instance, Bionicohand’s five-fingered bionic design exhibited rapid task execution and would have had a higher chance of winning if not for the “Haptic Bag” task, highlighting its advanced hand structure design. BionIT and Hannes Hand utilized a single motor to drive four fingers in parallel, maintaining high output force and practical features such as waterproofing and anthropomorphic design, showcasing greater maturity. MiaHand excelled in precision tasks such as pinch and tripod grasps, also maintaining a high level of maturity. While these teams demonstrated significant innovations in hand structure, the competition emphasized multidimensional performance, including multijoint control and sensory feedback. As previously noted, our team’s strengths in hand-and-wrist control along with intelligent sensory grasping provided a competitive edge, underscoring the importance of a holistic approach in prosthetic development.

In addition to the breakthroughs in key technical aspects, our comprehensive planning, meticulous preparation, and rigorous training were also critical to achieving this success. To summarize, the nontechnical factors contributing to the competitive success of Team HANDSON in CYBATHLON 2024 can be attributed as follows.

**Extensive competition experience.** The team members possessed substantial experience in participating series of CYBATHLON competitions. The HANDSON team’s R&D personnel previously served as Team Manager and Pilot in the CYBATHLON 2020 (Team Hands On, ranked 8th). In preparation for CYBATHLON 2024, team members also participated in the CYBATHLON Challenge 2023 and 2024 (Team SIBET ranked 1st and 3rd, respectively). Notably, Min Xu participated as a Pilot in two Challenges, gaining valuable competition experience. For other potential teams aspiring to prepare for CYBATHLON, it is essential that early preparation, adaptation to the competition environment, and active engagement in various stages of the Challenges are carried out.

**Right research purpose.** From the very beginning, the team was determined to complete all tasks. This rigorous standard enabled us to stand out in the intense competition. Among all the teams in CYBATHLON 2024, we were the only team to successfully complete “Haptic Bag”, which gave us a significant advantage on the field and exerted pressure on other teams. In fact, this success was a lesson learned from our experiences in previous Challenges. During the preparation for the 2024 Challenge, we chose to forgo “DIY”, while other teams successfully completed it, leading to us being surpassed by them. This served as a wake-up call, reminding us of the importance of thorough preparation.

**Flexible R&D strategy.** The tasks announced by CYBATHLON are highly challenging, even for well-established commercial prostheses, with examples such as the “Haptic Bag” and “Hanging Laundry” tasks. To enhance device performance during the competition, significant modifications to prosthetic designs may be required, which can be difficult for prostheses that have already reached a mature, finalized state. Our team tested various configuration schemes across two annual Challenges to identify the optimal approach. Ultimately, we adopted a flexible and suitable technical route, leveraging a gripper structure design developed by Team Hands On for CYBATHLON 2020 as a foundation, a design that had already been validated during the 2020 competition. Building on this, we integrated a camera, improved the wrist structure, and modified the fingertip shapes to better adapt the device to the specific requirements of this competition.

**Rigorous and meticulous training.** Treating every competition detail with precision during testing is of paramount importance. By training in Zurich in advance, we were able to account for the factor of the “competition environment”. Additionally, it was also important to attend the rehearsal in Zurich to help us identify the factor of the “referee”. For instance, as described in the section of “Result”, we made certain errors during the rehearsal day. By discussing the reasons for penalties with the referees, we were able to promptly rectify these mistakes, ensuring they were not repeated during the formal competition. To help our pilot remain confident during the event, we intentionally increased the difficulty of certain tasks during training sessions. For example, we tightened the lids in the “Container” task and reduced the diameter of the loops in the “Hot Wire” task to simulate heightened challenges.

**Close collaboration with patient.** During the development process, we incorporated feedback from the pilot, focusing on increasing device performance while optimizing the device weight to reduce user fatigue and discomfort. Through investigating the performance of other teams in previous competitions, we also communicated with the pilot to help understand the competitive level of other teams, enabling targeted training and improvement. Additionally, allocating sufficient training time for the pilot was crucial. This not only enhanced the pilot’s proficiency but allowed for identifying and solving potential reliability issues of the device through rigorous, high-intensity testing.

## 5. Conclusions

This paper highlights the innovative design and strategic approaches that led to the success of the HANDSON hand in CYBATHLON 2024. The HANDSON hand’s exceptional performance demonstrates the importance of integrating user-centered design with cutting-edge technology to meet real-world challenges. Moreover, the success underscores the value of early preparation, participation in challenges to refine strategies, and leveraging rehearsals to ensure compliance with competition rules. These efforts not only improved the device’s reliability and usability but provided critical insights for the advancement of multifunctional prosthetic hands. CYBATHLON sets a benchmark for future developments in assistive technologies, emphasizing the role of innovation and user needs in addressing the complex demands of daily living tasks for individuals with disabilities.

## Figures and Tables

**Figure 1 bioengineering-12-00228-f001:**
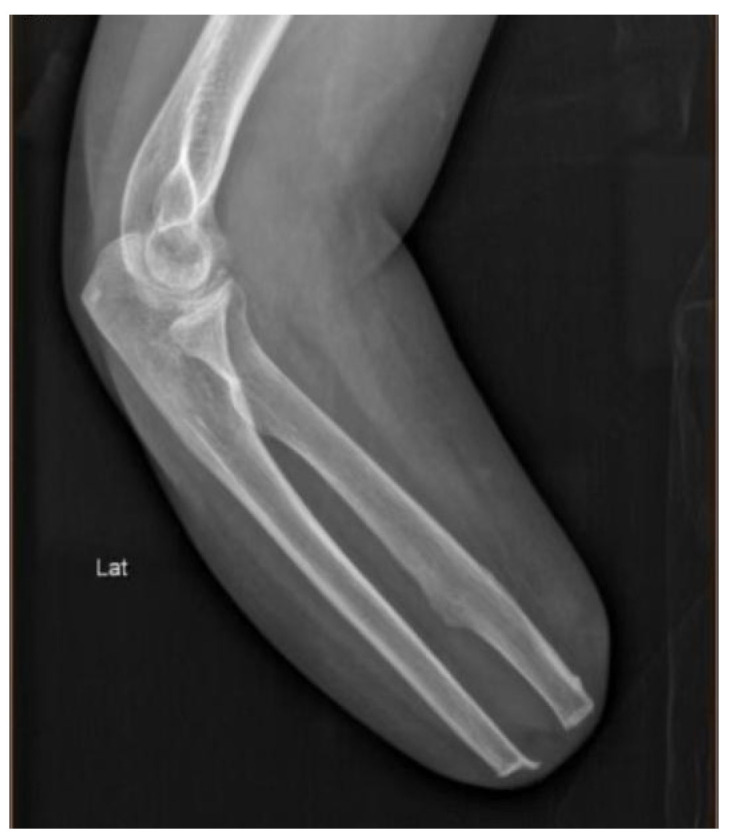
X-ray of forearm amputation (right hand), lateral view (lablled “Lat”).

**Figure 2 bioengineering-12-00228-f002:**
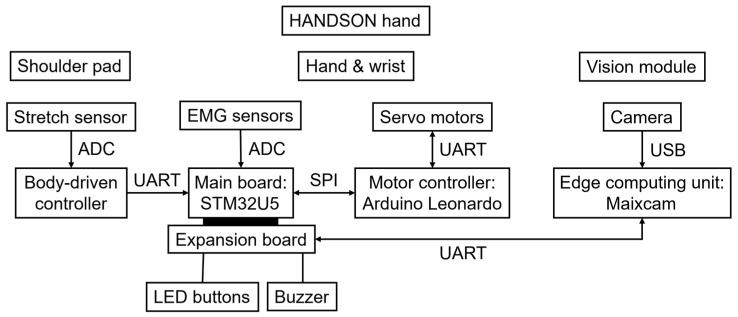
Diagram of connections between components of HANDSON hand in its current iteration. I2C, interintegrated circuit; SPI, Serial Peripheral Interface; UART, universal asynchronous receiver–transmitter; USB, Universal Serial Bus.

**Figure 3 bioengineering-12-00228-f003:**
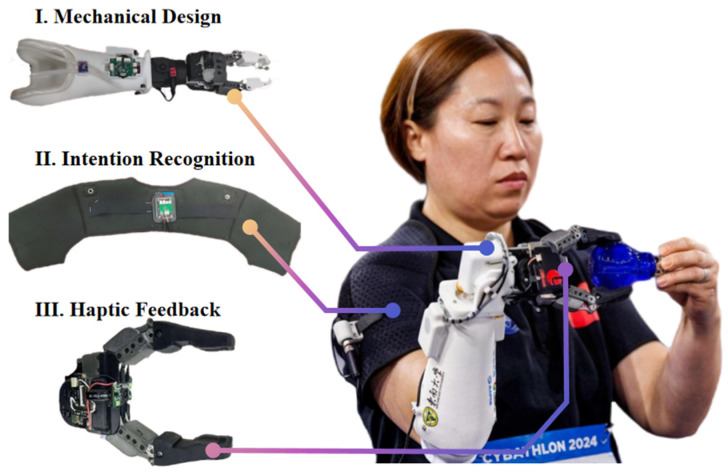
Key technology composition of the HANDSON hand.

**Figure 4 bioengineering-12-00228-f004:**
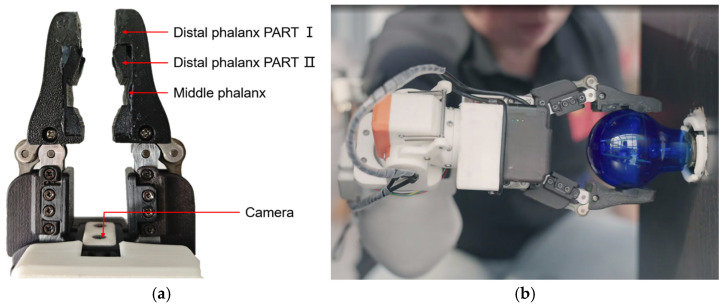
(**a**) Functional segmentation of finger grasping area. (**b**) Mechanical structure of the wrist.

**Figure 5 bioengineering-12-00228-f005:**
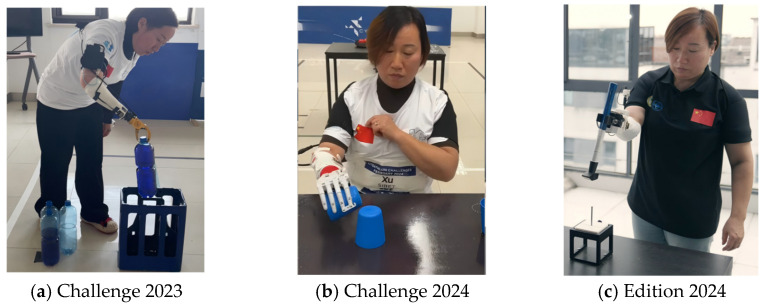
Pilot’s training in the laboratory for CYBATHLON.

**Figure 6 bioengineering-12-00228-f006:**
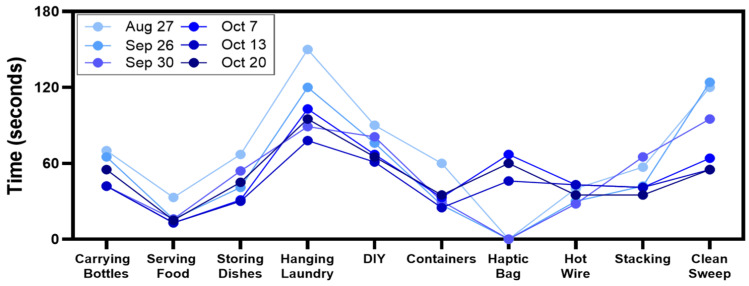
Performance improvement in prosthetic-assisted task completion over time.

**Figure 7 bioengineering-12-00228-f007:**
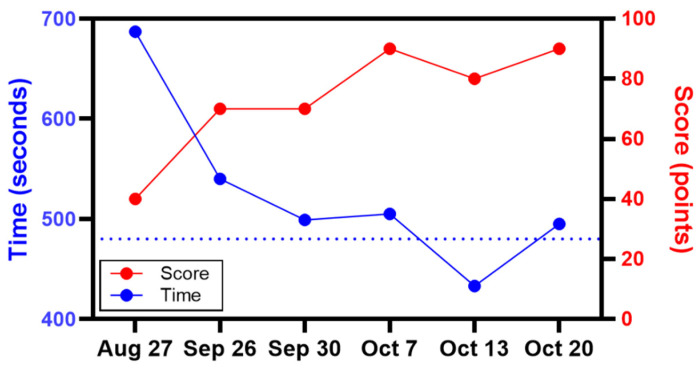
Progress in balancing task completion time and score optimization during training (The dotted line corresponds to the maximum allowed time, i.e. 480 seconds).

**Figure 8 bioengineering-12-00228-f008:**
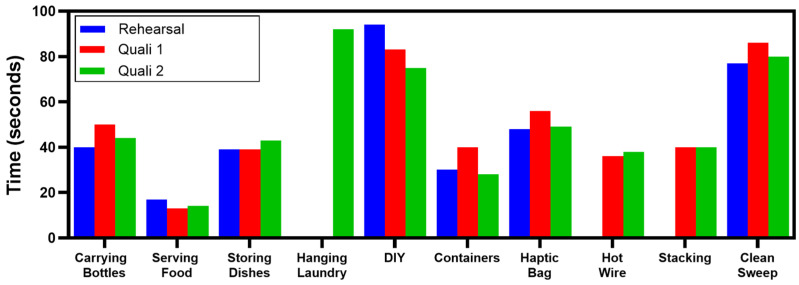
Performance of the HANDSON team across rehearsal and qualification rounds in CYBATHLON 2024 (tasks with 0 points are noted as 0 s. In “Rehearsal”, despite recording the completion time for the final task, the actual score was 0 because the total race time was exceeded).

**Figure 9 bioengineering-12-00228-f009:**
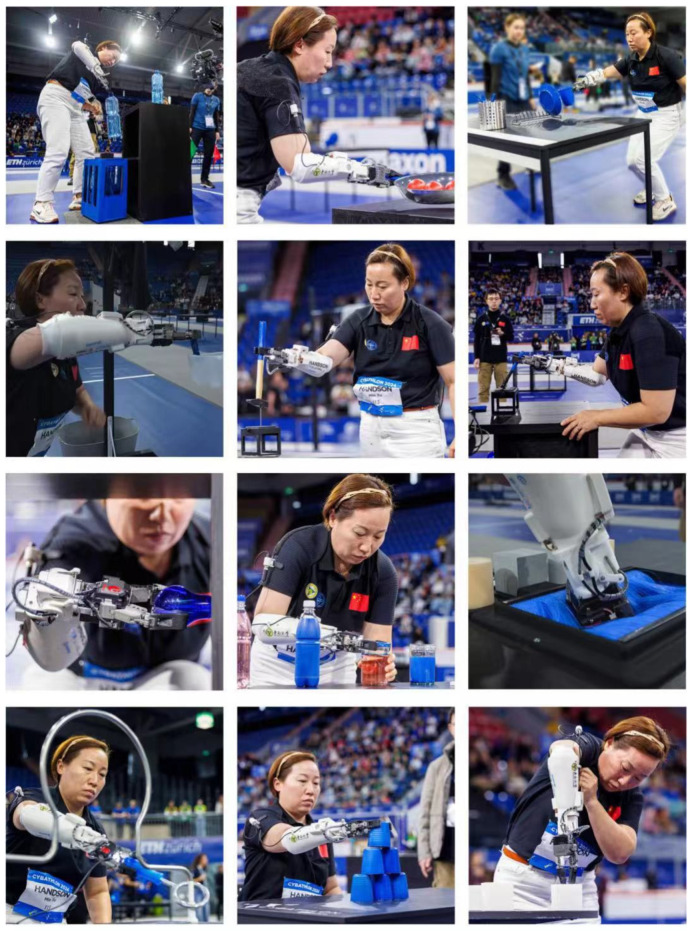
Snapshots of pilot from team HANDSON completing tasks during the CYBATHLON 2024 in Zurich (Since the DIY task included key subtasks, such as screwing in a light bulb and hammering a nail and then blowing it, the snapshots showcase several critical moments related to these subtasks).

**Figure 10 bioengineering-12-00228-f010:**
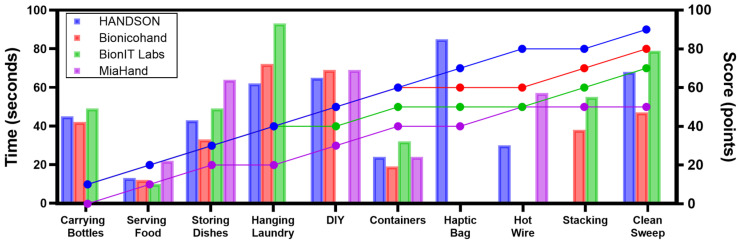
Completion times and score progression of four ARM teams in the finals of the CYBATHLON 2024.

**Table 1 bioengineering-12-00228-t001:** Comparison of race and rules between CYBATHLON 2020 and CYBATHLON 2024 (ARM Discipline) and objectives of rule modification.

Race and Rules for CYBATHLON 2020	Race and Rules for CYBATHLON 2024	Objective of Rule Modification
Allowed midtask replacement of prosthetic components	Not allowed	Enhancing the compatibility of prosthetic functionality
Allowed using the intact limb to assist prosthesis during object grasp	Not allowed (e.g., cutting off the power supply during heavy object lifting)	Prosthesis operational compatibility
/	NEW Task: Carrying Bottles	Grasping force, dexterity, fine motor control
/	NEW Task: Serving Food	Grasping force, bimanual coordination, fine motor control
/	NEW Task: Storing Dishes	Working range and control reliability
Task: Hanging Laundry	Must use prosthesis to fully close the zipper	Wearing convenience, bimanual coordination, control reliability
Task: Do-it-yourself	Add nail-pulling task, reduce space for screwing lightbulbs	Grip strength, wrist flexibility
Task: Containers	Spillage not allowed	Bimanual coordination, Grip strength
Task: Haptic Bag	Stricter restrictions	Perception accuracy
	NEW Task: Hot Wire	Wrist flexibility
Task: Stacking	Must remain seated	Wrist flexibility
Fine object manipulation	Difficulty remained the same	Dexterity

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
