# Peer review of "HANDSON Hand: Strategies and Approaches for Competitive Success at CYBATHLON 2024"

_bioengineering, 2025, doi:10.3390/bioengineering12030228_

Round 1
Reviewer 1 Report
Comments and Suggestions for Authors
I found the manuscript very interesting but for me it seems another kind of pubblication i.e. it contains contents that you could find on a website of the hand project, on the website of the competion or on a divulgative journal.
Author Response
Comment 1: I found the manuscript very interesting but for me it seems another kind of pubblication i.e. it contains contents that you could find on a website of the hand project, on the website of the competion or on a divulgative journal.
Response:Thank you for the comments. After carefully considering your feedback, we remain firmly convinced that our paper is more appropriately categorized as an "Article."
Firstly, our paper delves deeply into the academic research process and core technical details behind our team's participation in the competition. This is not a report on someone else's work, but a detailed narrative of our own research project, including the strategies, technologies, and insights gained throughout the competition. Such comprehensive coverage aligns more closely with the format and expectations of an "Article" than a "Communication."
Secondly, while limited information such as rankings and completion times may be publicly available online, our paper provides much more valuable insights into the crucial details that have not been disclosed elsewhere. These details, which constitute the core of our paper, include the technologies employed, competition strategies, and the underlying reasons for our achievements. This comprehensive explanation is essential for academic researchers seeking to understand and build their own work.
Thirdly, In response to feedback from other reviewers, we have already included more technical details to make our paper more appealing to academic researchers. These additions further strengthen our case for categorizing the paper as an "Article."
Based on the aforementioned reasons, we continue to believe that categorizing our paper as an "Article" is the most appropriate choice. We respectfully request the reviewer to reconsider this matter and acknowledge the unique academic value and comprehensive nature of our work.
Reviewer 2 Report
Comments and Suggestions for Authors
I would like to congratulate the research team on their success at CYBATHLON 2024 and thank them for an interesting paper. At the same time, I recommend that they fix some minor issues to improve their manuscript before it is published. The drawback is that the authors focus more on how they won the competition than on their design. At the same time, their design is the key result that other researchers will further use in their project. So the general recommendation is to make the paper more design-oriented rather than competition-oriented.
Other issues:
1) Please enlarge Figure 3 so that readers can see more details of your mechanical design. It is also possible to add a separate figure showing the parts of the solution that are not shown in Figure 4. Also, please add more labels showing the specific electronic parts that are demonstrated in Figure 2.
2) The caption of the figure for should be moved below the figure.
3) Please add details about the software of the prosthetic device. An additional figure with its structure is recommended.
4) Please demonstrate with an additional figure showing four types of objects that the visual recognition system was able to identify and how they looked on the camera.
5) Figure 8 is a bit misleading because the total time shown by the vertical stack makes no sense. In general, Rehearsal, Qualification 1, and 2 are separate trials. A recommendation is to stack these three events horizontally with three narrow bars of different color together for each separate task. A similar change should be made for Figure 10.
6) Please pay more attention in the Discussion section to specific technical solutions that contributed to the win. It would be great to see a direct comparison with the other teams' solutions. Which components of these solutions lead to success in specific tasks and which lead to failure? As I wrote in the beginning, currently authors pay too much attention to the competition part, forgetting that the main idea of this competition is to boast about the development of new better prostheses. This, additional analysis of the technical solutions is absolutely needed.
Author Response
Please switch to the attachment to see attached pictures.
Comment 1: I would like to congratulate the research team on their success at CYBATHLON 2024 and thank them for an interesting paper. At the same time, I recommend that they fix some minor issues to improve their manuscript before it is published. The drawback is that the authors focus more on how they won the competition than on their design. At the same time, their design is the key result that other researchers will further use in their project. So the general recommendation is to make the paper more design-oriented rather than competition-oriented.
Response: Thank you very much for your valuable and pertinent suggestions. In response, we have revised some content and increased the technical orientation to cater to the needs of academic and research readers. We have emphasized the design details of the prosthetic hand, enriched the hardware and software design, and ensured that the technical core stands out. At the same time, we have retained the necessary descriptions of the competition process to demonstrate the practical application effects of the design. To maintain a coherent and smooth structure of the article, the specific content related to these enhancements has been supplemented in the appendix.
Comment 2:Please enlarge Figure 3 so that readers can see more details of your mechanical design. It is also possible to add a separate figure showing the parts of the solution that are not shown in Figure 4. Also, please add more labels showing the specific electronic parts that are demonstrated in Figure 2.
Response: Thank you for your suggestion. We enlarged Figure 3, increasing its width from 9cm to 12cm, and added three additional figures that showing our solution in Appendix A. We have also included annotations and explanations of the prosthetic hand's structure and hardware details to provide readers with a clearer understanding of the mechanical construction of our hand.
“The HANDSON hand, from proximal to distal, consists of the socket, wrist joint, and palm joint(Figure A1). Inside the socket, two EMG sensors are embedded to collect electromyographic signals from user’s residual limb; externally, it is equipped with an STM32U5 main control board and a Maixcam vision module. The wrist joint includes a wrist motor that enables active rotation of the prosthetic wrist, as well as a mechanical trigger mechanism that allows for passive wrist flipping. The palm shown in Figure A3 features a single-DOF two-finger gripper with a parallel linkage mechanism, providing stable grasping capabilities.
Figure A2 displays the shoulder pad, which serves as a body-driven controller for the prosthetic hand. This device is equipped with two stretch sensors that capture the user's shoulder movement signals. These sensors are strategically placed to detect the motion and translate it into commands for the prosthetic hand, enabling the control of its grip opening and closing. By harnessing the natural movements of the shoulder, the shoulder pad provides a more intuitive and user-friendly interface for operating the prosthetic hand, enhancing the user's ability to perform daily tasks with greater ease and precision.
Figure A3 illustrates the structural design of the hand, highlighting key components that enable its functionality. The servo motor, integrated with a reduction gear, drives the opening and closing motions of the fingers via a connecting rod mechanism, ensuring precise control and force transmission. At the center of the palm, a camera is embedded, supported by a dedicated camera board, which transmit camera signals into USB protocol. This combination of electromechanical actuation (servo-driven finger movement) and embedded vision technology demonstrates a multifunctional approach to prosthetic hand design, balancing dexterity and sensory capabilities.”
Comment 3:The caption of the figure for (4?) should be moved below the figure.
Response: Thank you for pointing out this formatting issue. We have made the necessary corrections to Figure 4.
Comment 4:Please add details about the software of the prosthetic device. An additional figure with its structure is recommended.
Response: Thank you for your suggestion, we have drawn the corresponding flowchart and provided a more detailed introduction in the software design aspect. Please refer to the section Appendix A.
“The software framework of HANDSON hand operates through a structured workflow: initialization begins with startup and peripheral activation, followed by synchronized control signal acquisition via a 50Hz timer. Electromyography (EMG) sensor data is processed through an analog-to-digital converter (ADC), while stretch sensor feedback is collected via a serial communication interface. These inputs generate real-time control commands for wrist rotation and hand open/close actuation, complemented by grasping state recognition to monitor grip stability. Concurrently, a visual perception module identifies target object shapes and materials based on user commands, enabling selective recognition of specified targets through triggered inputs. Finally, the system provides audio-visual feedback based on both predicted object properties and real-time grasping outcomes, guiding user interaction with the prosthetic hand. ”
Comment 5: Please demonstrate with an additional figure showing four types of objects that the visual recognition system was able to identify and how they looked on the camera.
Response:Thank you for your suggestion. We have added additional images to demonstrate the four types of objects that the visual recognition system can identify in Line 645 Appendix A. Figure A5 shows the training set images we annotated for training the model, with four types of objects framed in the camera view using boxes of four different colors. Figure A6 shows the result image obtained by the visual recognition system when running the image detection model, with the four types of objects framed in green boxes and the recognition results of the object's shape and material displayed in the upper left corner. These results demonstrate that the system can accurately identify and locate these objects.
Comment 6: Figure 8 is a bit misleading because the total time shown by the vertical stack makes no sense. In general, Rehearsal, Qualification 1, and 2 are separate trials. A recommendation is to stack these three events horizontally with three narrow bars of different color together for each separate task. A similar change should be made for Figure 10.
Response: Thank you for your suggestion, we have adjusted the content of Figure 8 and Figure 10, changing the arrangement of the bars from vertical to horizontal layout.
Comment 7: Please pay more attention in the Discussion section to specific technical solutions that contributed to the win. It would be great to see a direct comparison with the other teams' solutions. Which components of these solutions lead to success in specific tasks and which lead to failure? As I wrote in the beginning, currently authors pay too much attention to the competition part, forgetting that the main idea of this competition is to boast about the development of new better prostheses. This, additional analysis of the technical solutions is absolutely needed.
Response:
Thank you for your suggestion, we have add more technical content in Line 480, section Discussion:
“At CYBATHLON 2024, numerous advancements in prosthetic technology were showcased, with several teams presenting innovative solutions that contributed to their success. Our team’s gripper-based design, while less dexterous compared to five-fingered bionic hands, demonstrated strong compatibility across various grasping modes and maintained competitive performance in precision tasks. By integrating wrist rotation and flipping capabilities within strict spatial and weight constraints, our device enabled faster task completion in activities such as rotating light bulbs and stacking cups. Notably, we achieved a breakthrough in the haptic bag task, not only being the first team to successfully tackle this task, but also laying a crucial hardware foundation for future advancements in embodied intelligence for prosthetic hands.
However, multi-joint active control in prosthetic devices does not necessarily translate to enhanced dexterity for users. Multi-joint simultaneous control based on myoelectric signals remains an area requiring deeper research, particularly in improving reliability [15, 33, 34]. Combining traditional cable-driven mechanisms with myoelectric interfaces as a supplementary control source is an intuitive idea[10]. However, traditional cable-driven methods are constrained by the rigidity of mechanical connections. For instance, when the shoulder is already in an extended range of motion (e.g., during Task 2, retrieving the casserole dish from the oven, which requires bending forward and reaching into the oven), the mechanical hand may unintentionally open and struggle to close. The result of this paper indicated that electrification of such cable driven system has improved its performance limits, demonstrating a better reliability and practicality across various tasks.”
Then we compare the key techniques among other teams, and summarize the advantanges of our team in Line 502:
“Other teams also showcased remarkable innovations. For instance, Bionicohand’s five-fingered bionic design exhibited rapid task execution and would have had a higher chance of winning if not for the haptic bag task, highlighting its advanced hand structure design. BionIT and Hannes Hand utilized a single motor to drive four fingers in parallel, maintaining high output force and practical features such as waterproofing and anthropomorphic design, showcasing greater maturity. MiaHand excelled in precision tasks like pinch and tripod grasps, also maintaining a high level of maturity. While these teams demonstrated significant innovations in hand structure, the competition emphasized multi-dimensional performance, including multi-joint control and sensory feedback. As previously noted, our team’s strengths in hand-and-wrist control along with intelligent sensory grasping provided a competitive edge, underscoring the importance of a holistic approach in prosthetic development.”

Round 2
Reviewer 1 Report
Comments and Suggestions for Authors
I confirm that the manuscript can not be published as a research article.
If MDPI wanted to publish it as another kind of pubblication such as a Case Study or a Divulgative article, I would be glad to approve it.
Comments on the Quality of English Languagenone